# Role of Employer Branding Dimensions on Employee Retention: Evidence from Educational Sector

**Noor Ul Hadi \*,† and Shahjehan Ahmed †**

Department of Leadership and Management Studies, National Defence University, Islamabad 44000, Pakistan; shahjehan.comsats@yahoo.com

**\*** Correspondence: noorulhadi@ndu.edu.pk or n_hadi1@yahoo.com; Tel.: +92-300-395-7833

† These authors contributed equally to this work.

**Abstract:** Owing to a "War for Talent" every organization is struggling for the best employer status. Thus, attracting, recruiting and retaining talented human capital is the primary focus of every organization. In this regard the aim of the present study is to answer the most frequent and recently asked question of what value(s) organization focus on the retention of their workforce? In so doing, Social Learning Theory and Reciprocity Theory were used as a theoretical background; to further proceed with the study, data was purposively collected from 204 respondents from educational institutes of Pakistan. Findings of the study revealed that development value has a substantial relationship with employee retention. Since, development of new knowledge and skills results in the improvement of one's present job. Furthermore, limitations and implications of the study are discussed.

**Keywords:** employer branding; employee retention; education sector; development value

## 1. Introduction and Background

Owing to the dynamic environment, among leading and competitive organizations, acquisition and retention of the best and qualified employees has become crucial for the survival of the organization and for gaining competitive advantage over other competitive organizations. It is evident from observations that human resources are the crown resource of the organization. Within a growingly saturated labor market employment and retention of premium human resources has become the core objective of businesses and employers (Collins 2001). This has started a talent acquisition war among organizations. It is very important that firms take steps to ensure talent acquisition and its retention to become competitive. Enterprises are focused on integrating the branding principles with the strategic human resource management strategies and practices. Branding is a concept derived from marketing, and employer branding is defined as the process of applying branding principles to Strategic HRM, which is used to distinguish firms from competing firms by inviting, inspiring and engaging both potential, as well as existing employees (Backhaus and Tikoo 2004). Employer branding is focused on a firm's individuality and uniqueness. A number of studies have concluded organization as a brand only relates to organizational characteristics that may include quality and innovativeness, etc. based on organizational values, culture, programs, and most importantly, its people. In current times, employer branding is being used as a new tactic to attract new, talented and qualified employees along with ensuring the loyalty of the existing ones. So, it is not only an employee seeking strategy, but also a strategy to ensure a trustworthy and appealing reputation of the organization (Nappa 2013).

Today's enterprises are investing a generous amount of resources and efforts to achieve the 'Best Employer' status due to the increased competition for recruiting and selecting new talented employees alongside competing for new customers and a higher market share (Berthon et al. 2005). The only

reason for this change in business trend is to be able to differentiate and gain competitive advantage over rivals (Lievens and Highhouse 2003). Employer branding directs the firm's operational practices through building a strong corporate image of the firm in the market and transforming it into an attractive workplace (Ahmad and Daud 2016). Today's organizations value their intellectual assets more than their hard assets and physical resources. Increasing levels of importance for recognition of the workforce, employee skills, employee knowledge and employee experiences are the sources of value for the firm and its stakeholders. This is why recruitment and retention of employees has become a major concern for organizations (Arachchige and Robertson 2013).

Due to the shift in recruitment trends of the businesses, it is crucial to study individuality, uniqueness and attractiveness in conjunction. As the organizations characteristically prefer to entice talent by developing an eye-catching employer image. Also, at the same time, guaranteeing that their built image is unswerving with employee's opinions of the identity of the organization (Lievens et al. 2007). There is still a great need for further research on employer branding to find valid answers to numerous queries. There also happens to be diminutive observed data as well as serious, autonomous studies on facets of employer branding. The aim of this study is to determine which dimension(s) of employer branding affects employee retention.

## 2. Rationale of Study

There is an ever-growing scope of possibilities for employers and employees in terms of employer branding. This leads to employer branding becoming a common practice among employers, making it a very well-known HRM practice of modern times. Further, previous studies on employer retention have exposed various related aspects mainly related to general management practices. There is little research available on the relationship between employer retention and employer branding. Thus, the topic has not been explored properly so far. This highlights the need for further research to be conducted on this topic, as there is limited existing literature available on the relationship between employer branding and employer retention. The current study will focus on further exploring the relationship among the two variables and will further add to the available knowledge base about employer branding and employee retention. The model is targeted to provide a standard for a better understanding of employer branding. It explains the aspects of employer branding which have an impact on employee retention. Not just that, it also provides an idea about gaining the loyalty of employees. To gain employees' felty to the organization, employer branding can be used to nurture the organization's skills and competencies. The findings of the study are also expected to provide a framework for employers to ensure efficient management and maintenance of a healthy employer brand. Further, it will also aid in providing a direction for future researchers to focus their research on core aspects of employer branding that are directly linked with the employee retention. Keeping the above aspects in mind, this study provides a sound rationale for researchers.

## 3. Significance

The current study incorporates insight about how employer branding can contribute towards making an organization more attractive and eye-catching for employees. This will provide an action-oriented outline for the enterprises in improving the organizational attractiveness to ensure the intake of new talented workers and successful retention of the existing employees. It will also give the managers and researchers both an enhanced understanding of the topic. Further it will elaborate the values that managers must focus on to retain their workforce in the real time business environment. The core objective of the research is to identify the exact dimension(s) of employer brand which are directly related to employee retention.

## 4. Parent Theories

The theoretical basis for the current study includes Social exchange theory and Reciprocity theory. Social exchange theory explains "social change and stability as a process of negotiated exchanges between parties" (Emerson 1976). In the business context, this theory is used to indicate a consensual, equally depending and gratifying process mainly involving transactions and/or simply exchange. For the current study, the social exchange theory provides the basis for proving the fact that in cases where an employer or and enterprise offers its employees value, proposition will result in higher levels of employee loyalty and faithfulness. This is directly related to the preservation of an organization's reputation, its future and its optimistic image, because of this social exchange between employer and employees, proving it to be a two-way thing. Additionally, Reciprocity theory also directly relates to the current study suggesting that "reciprocity is a social rule which implies that one should repay and individuals/people reward kind actions and punish the unkind ones" (Paese and Gilin 2000). In the context of the study at hand, within a business environment, if the enterprise (management) provides its employees with values, the employees will be morally bound to stay highly motivated and loyal to the firm, proving it to be a mutual process. Reaction to such organizational (management) actions improves the employee retention levels within the firm.

Both of the above-mentioned theories concurrently provide a sound base for the current study. Organizations are social entities and are formed only when the interaction occurs among the individuals, i.e., employer and employees. This interaction is mainly social in nature and is based on culture and formal and informal affiliations among employer and employees within the work setting. Based on the theory it is assumed that the organization must maintain a strong employer brand which can only be achieved through providing its employees with a strong sense of belonging through its core values, vision and mission. This results in increased levels of employees' commitment and loyalty. By doing so, employees become a great tool for the preservation of reputation and image in the market. This aspect is covered under social exchange theory, which is inter-linked with reciprocity theory because the social exchange among the employees involves actions and reactions on part of both parties, i.e., organization and its employees. Here it is important to note that firms with a sound organizational brand offers its employees strong culture, values and vision. Therefore, as per reciprocity theory in the organizations, which maintain a strong employer brand, employees repay with higher loyalty and commitment, thereby enabling firms to maintain higher levels of employee retention.

## 5. Review of Literature and Hypothesis Development

### 5.1. Employee Retention

As per Blattberg and Deighton (1996), in the marketing world it is usually advisable to secure your existing customers than to gain new ones. In the same theoretical context, the theory of employees' management states that it will be economical if an organization retains its current employees instead of replacing them or hiring new ones as the employer will have to invest more in the hiring process which includes writing job descriptions, posting positions, reviewing applications and then conducting interviews, not to mention hiring and then training the new employees. All this requires a vast amount of investment, efforts and time. Today there is a 'War for Talent' due to the urge of organizations to hire the best employees (Michaels et al. 2001). This implies a tougher competition between organizations as every other firm tries to attract better and skilled employees than its competitors, in addition to the retention of the existing ones (Alnıaçık and Alniacik 2012).

In existing research, employee retention has been defined as "a technique adopted by businesses to maintain an effective workforce and at the same time meet operational requirements" (Mehta et al. 2014). Another definition by Das and Baruah (2013) explains the concept as "a process in which the employees are encouraged to remain with the organization for the maximum period or until the completion of the project". The definition considered for the current research describes employee retention as, "a systematic effort to create and foster an environment that encourages

employees to remain employed by having policies and practices in place that address their diverse needs" (Workforce Planning for Wisconsin State Government 2005).

(Guthridge et al. 2008) stated that due to a significant increase in the shortage of talent globally, organizations these days tend to seek wide-ranging as well as comprehensive tactics and schemes that are certainly bound to entice and preserve probable as well as current employees. Preceding literature suggests that in the past, shortage of talent was caused by economic situations such as recession but now there is a probability of a shortage of talented employees in the future due to the demographic changes. Demographically, soon the existing generations of employees will retire and the younger/new generation of employees will take over. A major challenge faced by firms will be the retention of the new generation of employees because the younger generation has the tendency to switch their jobs (Lodberg 2011). Presently, employees have become opportunistic which means they are not confined to work with a single employer, due to a wide range of employment options available to them for switching their jobs, which is a challenge for the employers in the context of retention of the current workforce (Singh and Rokade 2014).

*5.2. Employer Branding*

The term 'Employer Branding' was first time conceptualized and defined by Ambler and Barrow (1996) as "the set of functional, economic and psychological assistances provided by employment, and recognized with the employing company". Employer branding consists of the following dimensions:

Interest Value: It assesses the degree of appeal of an employer who is responsible for providing a work situation with innovation and creativeness opportunities.

Social Value: It calculates the mark of appeal of an organization or a company providing a work environment with good and welcoming team spirit and decent respectable relations among coworkers.

Economic Value: It estimates the amount of attraction of an employer providing a worthy remuneration and profits.

Development Value: It is an attribute that estimates the degree of attractiveness of an employer providing career development.

Application Value: It determines the degree of attraction of an establishment providing the chance to exercise and train what is learnt.

In 2017, two more dimensions were added in the list by Dabirian et al., which are:

Management Value: It states that the good or bad influence of supervisors at work determines employee retention. It is more because of the bosses' attitude and their behavior towards their employees that the workers decide to stay in or leave a company, rather than because of the company itself. Good and bad supervisors influence employees tremendously. An employee's positive and negative experience with the boss also affects his/her social relationships.

Work/life Balance: It is an attribute determining that a proper balance among the employees' work and life allows them to work in harmony with all their identities. Employees should be considered more than just employees. It is important to consider that they have an identity outside the work. An appropriate balance between work and social life makes employees work more efficiently and effectively.

The employer branding concept was highlighted due to the growing competition among rival companies which required talent to compete and achieve growth and sustainability (Mosley 2007). In the 1990s, most business websites were static, i.e., they only posted advertisements in favor of their organization without any input from outside parties, but today's fast pace innovation in web-based technology and electronic media has allowed a two-way traffic, meaning that the employers and outside parties (employees, customer etc.) can also share their views related to the firm, its brands and offerings. This has a great impact on employer brands and in their ability to attract and maintain better (skills and abilities) employees (Ventura 2013).

Sullivan (2004) defined employer branding as a "targeted, long-term strategy to manage the awareness and perceptions of employees, potential employees, and related stakeholders with regards

to a particular firm". According to theoretical view point employer branding concept is described as the 'sum of a company's efforts to communicate to existing and prospective staff that the organization is a desirable place to work' (Lloyd 2002; Ewing et al. 2002). The term employer branding does not only refer to recruitment strategies that are short term and confined to job openings; employer branding is a long-term strategy which focuses on the continuous flow of innovative skills in the organization (Srivastava and Bhatnagar 2010). Employer branding basically consists of three stages:

i First is the designing of an attractive value proposition that includes the benefits that are to be offered to the future and actual workforce,

ii Second is the communication of the value proposition designed in the first stage,

iii Lastly implementation of the value proposition (Lievens 2007).

According to a research by Van Mossevelde (2010), employer branding is a contemporary administration priority in modern-day prominent companies which are said to be on the leading chart, and growing in importance. There are said to be five reasons for this:

Shortage of skilled labor:

Even though the business world faced an economic downturn in the years 2008 to 2011, which resulted in comparatively greater unemployment levels in a number of countries, the need for top talent and the right group of employees for the job still remains a crucial factor in today's employers' hiring practices.

More with less:

During the economic recession the economic burden was created for organizations, forcing them to reduce costs and increase their output. This made the need for the right people doing the right jobs even more precarious.

Growth and profitability:

It is important that organizations hire and maintain the finest possible employee talent as it proves to be vital for development and to uphold a competitive edge.

Popularity:

Research carried out into staff enrollment divulges that staff of all levels want to work for firms that are well known for their decent reputations. Unpopularity as an employer among employees can have a drastic consequence on the product and among corporate brands.

Strength:

Being attractive as an employer increases the influence of organizations for recollecting employees, which in some cases can also be regardless of salary levels.

Previous research also indicates that employer branding aids in the creation of organizational commitment. A sturdy employer brand increases organizational commitment levels once workers identify with their organization's values (Ind 2003). A respectable and durable employer brand encourages the workers to put in extra effort, which benefits the organization. Therefore, a maintained employer brand increases factors such as employee satisfaction, engagement and productivity efficiency (Xia and Yang 2010).

*5.3. Employer Branding and Employee Retention*

Employer branding helps in improving the recruitment of an organization, which helps to reduce recruitment costs. It also plays a vital role in improving a company's employee retention which results in reduced levels of employee turnover. It is essential that while developing a positive employer brand in organizations, managers should understand the importance of certain factors which play an important role in attracting possible employees to the firm (Alniacik et al. 2014).

Employer branding is a contemporary approach that is continuously expanding and it can keep up the firms' reputation in attracting and retaining employees (Ahmad and Daud 2016). Organizations are concerned with what their employees think about them as an employer. Employer branding not only results in employee loyalty and retention but also enhances the urge in people to work for that organization, therefore employees' word of mouth matters, whether it is negative or positive Dabirian et al. (2017). In the words of Suikkanen (2010), employer branding is an employer retention method because it is highly influential for the whole employment experience as it also encourages a good work environment while reducing voluntary turnover. According to Sokro (2012), an organization makes use of its employer branding strategy to attract employees in their business that can stay with the company and influence the choice of employees to stay or leave the organization. Employee loyalty and retention depends on the image of the organization that the employer has developed in the employee's mind, which is also important for attracting a new workforce (Dabirian et al. 2017).

Organizations with a strong brand image can acquire employees at comparatively low cost, improve employee relations, increase employee retention and offer lower pay scales as compared to its rivals (Riston 2002). Employer branding consists of three crucial benefits for the organization associated with recruiting, retaining and performance. The recruitment process is generally very expensive due to advertising, which only reaches a specific number of people. For a company with a strong employer brand, the recruitment process will generate a pool of talented employees without advertising costs. It is due to the strong positive reputation of the company as an employer that the majority of the capable and talented workers would look for job openings in that specific firm, which would lead to reduced cost of recruitment process. A strong employer brand will make the organization a desirable place for a talented work force and will attract potential employees. A strong employer brand will help in employee retention because the organization would be a desirable place to work and none of the employees would want to leave because every other organization will become less appealing to them. Also, if the organization is a desirable place to work, employees will enjoy working there which will increase their work performance (Taylor 2010). Most importantly as per Tanwar (2016), the dimensions of employer branding elucidate differences in levels of employees' organizational commitment and through it enterprise can enhance its employee retention levels.

By keeping the above discussion in view, one may establish that the existing work mainly focuses on exploring the positive association of employer branding and employee retention. Moreover, the existing literature advocates future exploration of the branding idea as a tool for employee retention. Here it is important to note that the existing studies have only targeted branding as a direct source of employee retention. There are no existing studies available on the direct relationship between employer branding and employee retention, which is the target of this study see Figure 1.

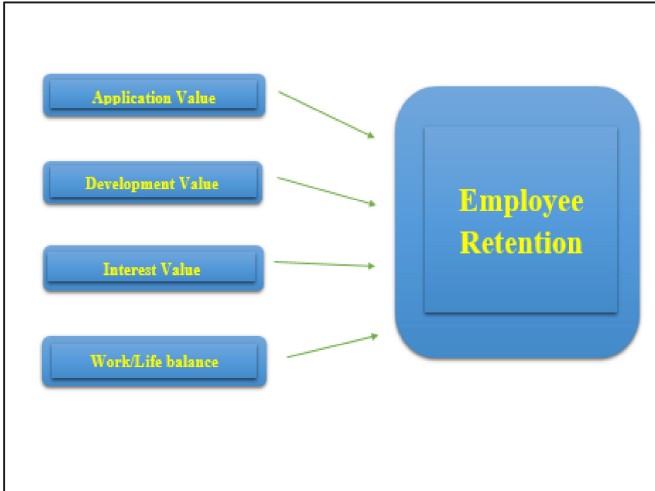

**Figure 1.** Dimensions of Employer Branding and Employee Retention.

*5.4. Hypotheses Development*

This section consists of the hypotheses for the proposed model:

**H1.** *There is a relationship between Application Value and Employee Retention.*

**H2.** *There is a relationship between Development Value and Employee Retention.*

**H3.** *There is a relationship between Interest Value and Employee Retention.*

**H4.** *There is a relationship between Work/Life Balance and Employee Retention.*

## 6. Research Methodology

*6.1. Instrument Development*

For this research, the questionnaire comprised of three sections which included demographics, employer branding and employer retention. There were a total of 27 items and a five point Likert scale was used (1–Strongly Disagree and 5–Strongly Agree) to measure the outcome and predictor variables.

*6.2. Employer Branding*

The scale of Berthon et al. (2005) was adapted for measuring the dimensions of employer brand with some modifications to set this scale according to Pakistani culture in order to make it easily understandable for employees. These dimensions included application value, development value and interest value. Questions related to work life balance were developed from the operationalization of work life balance by Dabirian et al. (2017) and a total of 16 items were used to measure employer brand.

*6.3. Employee Retention*

Questions related to employee retention were adapted by Kyndt et al. (2009) which consisted of 11 items and were used to measure employer retention.

*6.4. Sample and Data Collection*

This study has used quantitative research design to test the hypothesized relationship between the variables. The units of analysis were teachers and administrative staff members of branded institutes. The population consisted of educational institutes of Islamabad. The sample was selected by using judgment sample, since the scope of the study was limited to branded organizations and there are limited numbers of branded organizations in Pakistan, specifically Islamabad. The sample was collected from three universities in Islamabad, i.e., National Defense University (NDU), National University of Sciences and Technology (NUST) and COMSATS University. The selected educational institutes were represented by their teachers and administrative staff. The sample size was measured through power analysis. According to power analysis the sample size suggested was 204, and 123 valid responses were received out of 204 questionnaires, and the response rate is 60%.

*6.5. Data Analysis*

The data was analyzed by using the SPSS software version 20. Analysis was conducted via multiple regressions and the suitability of data for multiple regression was ensured. See Table 1.

**Table 1.** Descriptive Statistics.

| Demographics | Frequency | Percentage |
|---|---|---|
| **Gender:** | | |
| **Male** | 87 | 70.7 |
| **Female** | 36 | 29.3 |
| **Total:** | 123 | 100 |
| **Profession:** | | |
| **Teacher** | 77 | 62.6 |
| **Administrative Staff** | 46 | 37.4 |
| **Total:** | 123 | 100 |
| **Age:** | | |
| **20-30** | 42 | 34.1 |
| **31-40** | 48 | 39 |
| **41-50** | 24 | 19.5 |
| **51 & Above** | 9 | 7.3 |
| **Total:** | 123 | 100 |
| **Marital Status:** | | |
| **Single** | 41 | 33.3 |
| **Married** | 82 | 66.7 |
| **Total:** | 123 | 100 |
| **Qualification:** | | |
| **Matric** | 2 | 1.6 |
| **Intermediate** | 6 | 4.9 |
| **Bachelors** | 30 | 24.4 |
| **Masters** | 54 | 43.9 |
| **Doctorate** | 31 | 25.2 |
| **Total:** | 123 | 100 |
| **Work Experience:** | | |
| **1-5 years** | 43 | 35 |
| **5-10 years** | 33 | 26.8 |
| **10-15 years** | 20 | 16.3 |
| **15-20 years** | 10 | 8.1 |
| **20 & more** | 17 | 13.8 |
| **Total:** | **123** | **100** |

Source: Author's creation from SPSS results.

The descriptive statistics show that there were 70.7% males and 29.3% females. There were 62.6% teachers and 37.4% administrative staff who responded to the questionnaire survey. The minimum age group was 20–30, maximum age group were 50 and above, and 39% of the respondents lie in the 31–40 age bracket. The marital status shows that 33.3% were married and 66.7% were single. Among the respondents 1.6% had matric qualification, 4.9% of intermediate, 24.4% of bachelors, 43.9% of masters and 25.2% of Doctorate level. The work experience of respondents shows that 35% respondents had 1–5 years of experience, 26.8% of participants had 5–10 years of experience, 16.3% of respondents had 10–15 years of experience, 8.1% had 15–20 years of experience and 13.8% of respondents had 20 and more years of experience.

**Construct Validity**: To test construct validity that is convergent and discriminant, Hadi et al. (2016) proposed the evaluation of pattern and structured matrix.

*6.6. Exploratory Factor Analysis (EFA)*

According to Hadi et al. (2016), in Exploratory Factor Analysis (EFA) the different movements of observed variables are identified, considering cultural differences and research setting. Keeping this in mind, the author defined exploratory factor analysis (EFA) as "a statistical procedure used to reduce many observed variables to a small number of "factors/components".

The suitability of the data was ensured via KMO and Bartlett's test as shown in Table 2. Kaiser (1974) recommends a bare minimum KMO value of 0.5 and that the values between 0.5 and

0.7 are mediocre, the values between 0.7 and 0.8 are good, the values between 0.8 and 0.9 are great and the values 0.9 and above are superb as quoted by (Hadi et al. 2016). The results ruled that the value of KMO is 0.876, indicating that the data is fit for factor analysis.

**Table 2.** KMO and Bartlett's Test of Sphericity.

| KMO and Bartlett's Test | | |
|:---|:---|:---|
| Kaiser-Meyer-Olkin Measure of Sampling Adequacy. | | 0.876 |
| Bartlett's Test of Sphericity | Approx. Chi-Square | 1376.417 |
| | Df | 120 |
| | Sig. | 0 |

Source: SPSS results.

Further, Bartlett's Test of Sphericity tells the strength of the relationship aimed at measuring the multivariate normality of set of distribution. At the same time, it also checks the null hypothesis that the original correlation matrix is an identity matrix. If the significance value is less than 0.05 then that means the data has not produced an identity matrix and are thus approximately multivariate normal and acceptable for further analysis (Pallant 2013).

It is further discovered that 73.5% of the variance is explained by application value, development value, interest value and work/life balance.

According to the above pattern matrix (Table 3), all of the values are greater than 0.5, which means that all the items are converged on for their respective factors and convergent validity is ensured. If the value for Cronbach alpha coefficient is greater than 0.60 then it lies within the acceptable range (Nunnally and Bernstein 1994). Discriminant validity was also ensured via inspecting structure matrix. After construct validity, the reliability in the scales were checked. In the above table the Cronbach alpha exceeded the minimum acceptability range of 0.60 as suggested by Nunnally and Bernstein (1994), therefore indicating that all the items used in the constructs are reliable and consistent.

**Table 3.** Pattern Matrix.

| Items | Component | | | | |
|:---|:---:|:---:|:---:|:---:|:---:|
| | **1** | **2** | **3** | **4** | **α** |
| **Developmental Value:** | | | | | **0.9** |
| Our organization provides foundation for future employment | 0.863 | | | | |
| Employees feel good about themselves as a result of working for our organization | 0.802 | | | | |
| Employees feel more confident as a result of working for our organization | 0.724 | | | | |
| Our organization provides experience that improves our career | 0.719 | | | | |
| In our organization there is appreciation from management towards employees | 0.504 | | | | |
| **Interest Value:** | | | | | **0.88** |
| Our organization produces original products and services | | 0.78 | | | |
| Our organization provides unique work practices | | 0.77 | | | |
| Our organization provides an exciting work environment | | 0.73 | | | |
| Our organization uses values and makes use of our creativity | | 0.71 | | | |
| Our organization produces high-quality products and services | | 0.7 | | | |
| **Work/Life Balance:** | | | | | **0.86** |
| Our organization offers flexible work arrangements for employees | | | 0.92 | | |
| Our organization looks after employees' work/life balance | | | 0.79 | | |
| Flexible work arrangements offered by our organization enable us to be successful on and off the job | | | 0.73 | | |
| **Application Value:** | | | | | **0.8** |
| Our organization provides an opportunity to our employees to teach others what you have learned | | | | 0.77 | |
| Our organization provides an opportunity to our employees to apply what you have learned | | | | 0.63 | |
| Our organization is application oriented | | | | 0.52 | |

Source: Author's creation from SPSS results.

## 7. Multiple Regression Analysis

### 7.1. Assumptions of Multiple Regression

Normality in the model was ensured via histogram, whereas linearity was ensured through pp-plot. According to Hutcheson and Sofroniou (1999), Durbin–Watson should lie between the two critical values of 1.5 < d < 2.5. Durbin–Watson d = 2.077, therefore we can assume that there is no first order linear auto-correlation in our multiple linear regression data. Values of Variance Inflation Factor (VIF) should be less than 4 Steenkamp and Van Trijp (1991) and all the values in the table are less than 4, hence, it indicates that there is no multicollinearity problem among the dimensions of the predictor variable see for detail Table 4.

**Table 4.** Multiple Regression Assumptions.

| Assumptions | Threshold Point | References | Model Values |
|---|---|---|---|
| Durbin Watson | 1.5–2.5 | (Hutcheson and Sofroniou 1999) | 2.077 |
| Variance Inflation Factor (VIF) | VIF < 4 | (Steenkamp and Van Trijp 1991) | **IV:** 1.935 **DV:** 2.783 **AV:** 2.384 **WLB:** 2.032 |
| Mahal Distance | Critical Value (values above the critical value are outliers) | (Algur and Biradar 2017) | **Min:** 0.217 **Max:** 21.451 **Mean:** 3.967 |
| Cooks Distance | 4/n | (Algur and Biradar 2017) | **Min:** 0.000 **Max:** 0.139 **Mean:** 0.010 |
| Normality Linearity Homoscedasity | | | Histogram PP-Plot **Scatter Plot** |

Source: Author's creation from SPSS results.

From the above test it is evident that the development value is linked to the dimension of employer branding with the employee retention in branded institutions of Pakistan especially in Islamabad with a beta of 21.4% and *t*-value of 2.47 with *p*-value 0.15 as it met the rule of thumb where p-value is less than 0.05. However, the rest of the dimensions are not confirmed as a key dimension in this setting.

The multiple regression model summary reports that the coefficient determination R square equals 0.305, which means that 30.5% of the variation in the dependent variable (employee retention) can be explained by all the independent variables in the research see for detail Table 5.

**Table 5.** Regression Analysis.

| Hypothesis | Path | Beta | Std. Error | t Statistic | P Value | Decision |
|---|---|---|---|---|---|---|
| **H1** | AV – ER | 0.09 | 0.075 | 1.208 | 0.23 | Rejected |
| **H2** | DV – ER | 0.214 | 0.086 | 2.47 | 0.015 | Accepted |
| **H3** | IV – ER | 0.045 | 0.065 | 0.692 | 0.49 | Rejected |
| **H4** | WLB – ER | 0.053 | 0.064 | 0.823 | 0.412 | Rejected |
| $R^2$ | 0.305 | | | | | |
| Adjusted $R^2$ | 0.281 | | | | | |
| F-Value | 12.931 | | | | | |
| Sig | 0.0000 | | | | | |

Source: Author's creation from SPSS results.

### 7.2. Discussion on Findings

Employer branding is considered strategically crucial to gain employee retention. Employer branding enables the employer to build an image in people's mind that the firm is a great place to work. According to this study, out of four only one dimension has a significant

relationship with employee retention. There is a significant positive relationship in development value and employee retention but there is no relationship between application value, interest value, work/life balance and employee retention. There are currently eight dimensions of employer brand and we have only taken four in consideration. Maybe the combined effect of other relevant dimensions would yield different results.

This finding supports preceding research that development value is a crucial value in employee retention. Ahmad and Daud (2016) suggested that due to an intense extent of competition among candidates these days, most of the employees are loyal to an organization that can ensure employee development programs as that is a certain way to ensure bright future opportunities. It is common human nature that praise affects the behavioral reactions of human beings. Hence, more the employees are appreciated due to their work, the more employees feel good about them and feel confident while working for that organization and that they would retain in the organization.

Previous findings have also highlighted the importance of training and development as a crucial tool for employee retention and commitment (Newman et al. 2011). In today's turbulent environment, employees prefer the employers who equip them with latest knowledge and skills. Arachchige and Robertson (2013) added to this theory suggesting that attributes important to individuals are individual growth through the provision of numerous career opportunities and job security as well.

The Higher Education Commission (HEC) of Pakistan has introduced a Model Track Process to specify the rules and regulations for the implementation and execution of the tenure track process at the degree awarding institutes and Universities of Pakistan.[1] Keeping this in mind, the study has revealed that in the higher education sector of Pakistan there is a strong relationship between development value and employee retention. The faculties serving at the Universities/Degree awarding institutions prefer to stay with the institution following the Tenure Tracking System (TTS) program. This is because the program is designed to offer the faculty member a systematic framework directed towards career growth opportunities and provides them with the development value they seek from the employer. Pakistan's labor market has alarming levels of skills and talent shortages, employees usually prefer the organizations which can help them to enhance their career path by providing them with opportunities along with a clear path through which employees may work on their development and career enhancement.

Previously, five dimensions of employer brand were defined which consisted of interest value, application value, development value, social value and economic value. Later in 2017, two values i.e., management value and work/life balance were added in the dimensions of employer brand. In the similar year 2017, diversity value was another value added to the dimensions of employer brand. Until now, there are eight dimensions of employer brand and further we can recommend a ninth dimension that can be added to employer branding is Psychological value. Psychological value is a value which is related to the freedom of movement of an employee with no fear or risk of life etc. For example, if a person works in an International Non-Governmental Organization (INGO) and he is posted to Islamabad office and he visits Federally Administered Tribal Area (FATA). In case Taliban kidnaps him and categorizes him as an American citizen, his life will be in danger. Hence, employees will prefer to work for an organization where they will have lesser fear and risk of life.

---

[1] Due to the variance in the management structure Higher education institutions of Pakistan, TTS aim is to improve the quality of the faculty serving the higher education needs of the country. Additionally, involved institutions are allowed to modify or alter these model tenure track statutes as per their needs after consulting the HEC. Considering that the alteration must not change the fundamental spirit of the tenure track process of an open recognition of merit that is employment opportunity should only be offered to individuals based on excellence as per the set criteria in the relevant subject matter. Further, the institutions following TTS become eligible to receive additional Government funding for tenure track appointments. This is a significant development in the higher education sector of the country as it will have a dual effect on the overall higher education system of the country in context of the quality and volume of the new intellect that will be mentoring the future generations of the nation.

The current findings reassert the concept given by social exchange theory and reciprocity theory, even though three of the four dimensions of the employer branding under investigation do not show any direct relationship. These dimensions certainly play a significant role in employee retention indirectly through effecting employee satisfaction and commitment as per existing research, whereas the developmental value plays a direct role in enhancement of the employee retention levels in an organization which maintains a sound employer brand. The theoretical basis for the current study promotes the idea of social exchange among employers and employees, due to which organizations who offer developmental value tend to motivate their employees to stay with them for longer durations more effectively. This is due to the fact that employees feel more obliged to the organization and repay through staying committed to the firm.

## 8. Conclusions

The scope of employer brand is constantly expanding and employee retention is an important aspect in the success of every organization. From this study we can therefore conclude that development value is a crucial value for employees as it enhances employer retention while the other values have no significant relationship with employee retention. Employees tend to leave the organization if they are not praised accordingly or if the management of the organization exhibits excessive and unnecessary complaint factor. Findings of this study will help employers and managers to find out on what to focus more in order to retain workforce in public sector organizations. Organizations need to pay more attention towards the development aspect of employees. As appreciation is a basic part of development value, an employer should never underestimate the power of praising his employees to certify an improvement in the working skills. Apart from that, another tactic to gain better employee performance is to equip the employees with necessary knowledge and good skills for the performance of a given job. Doing so will be beneficial for both the parties.

Further, since in the study data was mainly collected from public sector organizations and the results indicated the developmental value to be a major factor playing a vital role in the employee retention. Further the remaining three dimensions may not have shown any direct relationship with the employee retention but as per existing work they do play an important role in enhancing the retention levels of the organization through improved satisfaction and commitment. These results or findings may differ in private sector organization as it may highlight other factors as the driving force behind the employee retention in the private sector organizations. Therefore, the generalization of the results in public and private sector will not be possible.

## 9. Implications

Development value is a necessary and crucial value for retention of employees. Managers should prioritize recognition and appreciation among the employees for better performance and retention. Managers need to develop necessary and updated skills among employees which would help the organization to take the benefit of their employee's maximum potential and will also help employees to develop themselves more effectively, so that they can opt for better opportunities within or outside their current organization. If managers appreciate the work done by the employees and provide them with development value, this would result in employee loyalty and will lead to employee retention.

Employers' focus must be on the development of their workforce for maximizing the firm's output and to keep their employees loyal to the organization. This is because today's employees are most concerned with their development. This will not only enable the employers to retain their employees but also will allow them to attract new talent efficiently. This can be achieved through diverse development opportunities involving job specific training programs through which employees can enhance their knowledge base about the job at hand. This will allow the employer to retain the knowledge workers through the development value offered by the employer. As literature highlighted at many occasions that career-oriented people tend to switch their jobs more often if they are provided with better opportunity. Employees are usually concerned with self-development and perceive those

organizations as good employer which continuously develops its employees. Development value offered by an employer is a critical factor for the enhancement of the firm's image as a great employer to work for in the market.

Employees can also benefit through this research. Due to the dynamic environment and skills and talent shortages, they should look to work for the organizations which provide them with personal development opportunities. Employees who tend to develop themselves can easily work in a dynamic environment and are employable. If employees can develop themselves they won't be reliant on a single employer, they can work for wide range of employers in different industries. This will provide them to unlimited employment opportunities from the market.

## 10. Limitations

There were originally eight values defined in the previous research theoretically. However, this paper focused on four values instead. The sample that has been used in this paper is judgement sample due to the presence of limited number of branded organizations in Pakistan. Findings of this paper are also limited to educational institutes. The sample size was 204 but valid responses were received from only 123 respondents, but the results could have been much better with a larger sample size.

**Author Contributions:** All the authors contributes equally.

**Funding:** This research received no external fundings.

**Conflicts of Interest:** The authors declare no conflict of interest.

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
