# Peer review of "Role of Employer Branding Dimensions on Employee Retention: Evidence from Educational Sector"

_admsci, doi:10.3390/admsci8030044_

Round 1

Reviewer 1 Report

The paper is based on an interesting concept of employer branding which is spreading fast as an emerging agenda on top table of foreign organizations. The introduction needs to provide more critical information regarding the emergence of employer branding per se and mus highlight key references/sources that will underpin the study. For example, the author(s) have not provided a single source to back up their rationale. This is one of the most important sections of any study and until and unlessthe rationale is clearly outlined, the significance of the study cannot be established. Hence, this section needs particular attention. Furthermore, a clear connecion between the current study and the underlying thoeries mus be made. The author(s) have merely highlighted the undeying theories while not connecting them strongly enough with the study.

The literaure review tends to overly describe/discuss the employer branding concept. The author(s) must make the review progressive in nature and must move on swifly from defining the concept to developing it in their study rather critically and clearly demonstrating its application and use in the current study.

Regarding methodology, the response (questionnaires) do not seem sufficient to develop/confirm concrete theories. A sample of three universities should definitely have more respondents for testing of hpotheses.

Discussion needs further critical insight in terms of the relationship between employer branding and employee retention.

Overall, its a very simplitic paper and certainly needs more elaboration in terms of the discovered relationship between employer branding and employee retention. It also needs to be more criticl in nature and establish theortical connections more concretely.

Author Response

Respected Sir,

Find attached file for your kind review.

Round 2

Reviewer 1 Report

Authors have addressed majority of my comments and concerns raise in the first draft.

Author Response

Respected Sir,

Kindly find attached change sheet for your kind review.
